# Characterization of Preoperative, Postsurgical, Acute and Chronic Pain in High Risk Breast Cancer Patients

**DOI:** 10.3390/jcm9123831

**Published:** 2020-11-26

**Authors:** Patrice Forget, Taalke M. Sitter, Rosemary J. Hollick, Diane Dixon, Aline van Maanen, Alain Dekleermaker, Francois P. Duhoux, Marc De Kock, Martine Berliere

**Affiliations:** 1Institute of Applied Health Sciences, Epidemiology Group, School of Medicine, Medical Sciences and Nutrition, University of Aberdeen, Aberdeen AB25 2ZD, UK; tsitter@uni-bremen.de (T.M.S.); rhollick@abdn.ac.uk (R.J.H.); 2Department of Anaesthesia, NHS Grampian, Aberdeen AB25 2ZD, UK; 3Aberdeen Centre for Arthritis and Musculoskeletal Health, University of Aberdeen, Aberdeen AB25 2ZD, UK; 4MRC versus Arthritis Centre for Musculoskeletal Health and Work, Aberdeen AB25 2ZD, UK; 5Institute of Applied Health Sciences, Health Psychology, School of Medicine, Medical Sciences and Nutrition, University of Aberdeen, Aberdeen AB25 2ZD, UK; diane.dixon@abdn.ac.uk; 6Biostatistics Unit, King Albert II Institute, Cliniques universitaires Saint-Luc, 1200 Brussels, Belgium; Aline.vanmaanen@uclouvain.be; 7Clinical Pharmacology Unit, Cliniques universitaires Saint-Luc, 1200 Brussels, Belgium; Alain2000@live.be; 8Institut Roi Albert II, Service d’Oncologie Médicale, Cliniques universitaires Saint-Luc and Institut de Recherche Expérimentale et Clinique (POLE MIRO), UCLouvain, 1200 Brussels, Belgium; Francois.duhoux@uclouvain.be; 9Department of Anesthesiology, Centre Hospitalier Wallonie Picarde (CHWAPI), 7500 Tournai, Belgium; Marc.dekock@chwapi.be; 10Department of Gynecology, Breast Clinic, King Albert II Institute, Cliniques universitaires Saint-Luc, UCLouvain, 1200 Brussels, Belgium; Martine.berliere@uclouvain.be

**Keywords:** ketorolac, breast cancer, acute pain, chronic pain, musculoskeletal pain

## Abstract

Background: Pain after breast cancer surgery remains largely unexplained and inconsistently quantified. This study aims to describe the perioperative pain patterns in patients with breast cancer, up to two years after surgery. Methods: This is a pre-planned sub-study of the Ketorolac in Breast Cancer (KBC) trial. The KBC trial was a multicentre, prospective, double-blind, placebo-controlled, randomised trial of a single dose of 30 mg of ketorolac just before breast cancer surgery, aiming to test its effect on recurrences. This sub-study focuses only on pain outcomes. From 2013 to 2015, 203 patients were randomised to ketorolac (*n* = 96) or placebo (*n* = 107). Structured questionnaires were delivered by telephone after one and two years, exploring the presence, location, permanence, and frequency of pain. Patients’ perceptions of pain were captured by an open-ended question, the responses to which were coded and classified using hierarchical clustering. Results: There was no difference in pain between the ketorolac and the placebo group. The reported incidence of permanent pain was 67% and 45% at one and two years, respectively. The largest category was musculoskeletal pain. Permanent pain was mainly described in patients with musculoskeletal pain. The description of pain changed in most patients during the second postoperative year, i.e., moved from one category to another (no pain, permanent, or non-permanent pain, but also, the localisation). This phenomenon includes patients without pain at one year. Conclusions: Pain is a complex phenomenon, but also a fragile and unstable endpoint. Pain after breast cancer surgery does not necessarily mean breast pain but also musculoskeletal and other pains. The permanence of pain and the pain phenotype can change over time.

## 1. Introduction

Pain remains a significant problem after breast cancer surgery. The problem is largely unexplained and inconsistently quantified [1]. The dynamics of acute and persistent postoperative pain are variably reported, with mixed results. Even the localisation of the pain is reported variably, leading to a potential misunderstanding when examining persistent pain after breast cancer surgery.

This may be partly due to the fact that some of the pain syndromes are associated with adjuvant treatment, such as polyneuropathy (induced by chemotherapy) and musculoskeletal pain (caused by endocrine therapy), and do not meet the definition of postsurgical pain. However, other pain syndromes are mostly unexplained, even if they are, at least partially, related to surgery, such as shoulder pain [2].

From the patient’s perspective, pain after treatment for breast cancer can affect quality of life, whether or not it is clearly related to the surgical insult. In addition, as the cause of the onset of new musculoskeletal pain is usually not explained, it would be important to describe postoperative pain patterns, not only limited to the surgical site, but also in other areas.

This sub-study of the Ketorolac in Breast Cancer (KBC) study aims to describe the perioperative pain patterns in patients with breast cancer, up to two years after surgery. The KBC trial was a randomised, placebo-controlled trial designed to test the hypothesis that a single intraoperative dose of ketorolac may be associated with prolonged disease-free survival after surgery in breast cancer patients (NCT01806259) [3]. The trial reported no effect on acute postoperative pain from a single dose of ketorolac. Acute pain was considered a safety outcome, but a sub-study on persistent pain after surgery was pre-planned. Data were therefore collected prospectively, capturing evidence on perioperative pain and pain up to two years after surgery.

## 2. Patients and Methods

### 2.1. Settings

This is a sub-study of a multicentre, prospective, double-blind, placebo-controlled, randomised phase III trial of perioperative ketorolac in patients with high-risk breast cancer. Patients were randomised in a 1:1 ratio to receive either 30 mg of ketorolac tromethamine (Taradyl, N.V. Roche S.A., 1070 Brussels, Belgium) or placebo before the surgical incision. The present study not only focuses on comparing pain outcomes between groups, but more broadly describes the characteristics of acute and persistent pain across the cohort. The study was approved by the institutional review committees of all the participating centres (central ethics committee: Catholic University of Louvain, EUDRACT 2012-003774-76) and was conducted in accordance with the Declaration of Helsinki and applicable national and European laws. Patients provided written informed consent. This report is written in accordance with the CONSORT directives [4].

### 2.2. Participants

Patients provided written informed consent for the main trial as well as the pain sub-study. Inclusion criteria included: patients scheduled for curative breast cancer surgery, weighing between 50 and 100 kg, and aged 18 to 75, with a neutrophil-to-lymphocyte ratio > 4 and/or showing evidence of lymph node invasion and/or triple negative status. Patients were excluded if they refused or could not follow part of the protocol (preoperative screening, complete treatment, or follow-up), in the case of history of cancer in the last 2 years, planned non-curative surgery (confirmed preoperative T4 or M1), and in the presence of a contraindication to ketorolac (or to another drug used in the standardised anaesthetic protocol).

### 2.3. Intervention

General anaesthesia followed a standardised protocol including a continuous infusion of propofol, ketamine 0.3 mg/kg, clonidine as needed to maintain hemodynamic stability up to 4 µg/kg, and sufentanil per bolus of 0.1 µg/kg if necessary. Postoperative analgesia included paracetamol as needed (3 to 4 g/day), tramadol 50 mg, and piritramide 10 mg IM/6h in the case of severe pain.

### 2.4. Data Collection

Between February 2013 and July 2015, 203 patients from four sites in Belgium were randomised to ketorolac (*n* = 96) or placebo (*n* = 107). Oncological follow-up was carried out by the oncologist in charge of the patients, initially every 3 months after surgery and then, every 6 months after the second postoperative year.

In addition, structured questionnaires were delivered by telephone by a research coordinator (Alain Dekleermaker) after one and two years. These explored the presence of pain, its location, permanence, frequency, and other characteristics, other sensations and their location, and the presence of a phantom chest sensation (additional material, Case Report Form—CRF 7 and 16). Musculoskeletal pain (arthralgia, myalgia, low back pain), neuropathy, headaches, and other types of pain were also noted. Patient perceptions of pain were captured by an open-ended question, by specifically asking to “describe the pain” (if the pain was not localised in the initial surgery site/the arm).

### 2.5. Data Processing and Analyses

#### 2.5.1. Quantitative Analyses

The quantitative analyses have been described previously [3]. In short, the study was designed to detect a 33% reduction in the risk of recurrence, with a power of 0.8 and an alpha of 0.05. The intention-to-treat population was used for all efficacy analyses of a possible effect of ketorolac. Patient demographics, baseline characteristics, and oncology treatments were summarised using descriptive statistics (mean, standard deviation, median, minimum and maximum values) for continuous parameters, and frequencies and percentages for categorical data. Group comparisons were made using the Mann–Whitney, Fisher, or Chi-square test as appropriate, and 95% confidence intervals were provided where required. All statistical tests were two-sided. For several comparisons, a Bonferroni correction was used, based on a Type I error of 5%. All data were collected using REDCap (Research Electronic Data Capture System, Vanderbilt University, Nashville TN 37235, USA) [5,6] and analysed using SAS statistical software version 9.4 (Copyright, SAS Institute Inc.) and SPSS for Windows, version 25.0 (IBM Corp. Released 2017, Armonk, NY, USA). The graphs were constructed using SankeyMATIC (http://sankeymatic.com) and STATISTICA (data analysis software system) version 7 (StatSoft Inc. 2004, Tulsa, OK, USA).

#### 2.5.2. Thematical/Framework Type Analyses

The open question from the telephone questionnaires was transcribed verbatim and analysed by Patrice Forget (PF) and Taalke Sitter (TS) as follows:(1)Identify information about the type of pain, the site and cause, and how it has changed over time.(2)Organise the codes according to the similar meaning identified in a second and third round until the saturation is considered reached. The codes were grouped into categories, without any interpretation concerning a possible aetiology: “permanent pain”, “musculoskeletal pain”, “pain located in surgical area”, “other located pain”, “polyneuropathy”, and “no pain”. “Permanent pain” refers to “pain all the time”; “musculoskeletal pain” refers to any pain originating from or resembling arthralgia, myalgia, or low back pain; “pain located in surgical area” refers to the patient’s interpretation, primarily related to the pain localisation; “polyneuropathy” refers to pain in the extremities associated with positive and/or negative symptoms (like paraesthesia and numbness).(3)Classify and group codes in clusters: patients with different types of pain (e.g., with both polyneuropathy and musculoskeletal pain) were counted in different categories. In contrast, patients experiencing more than one type of pain within a category were counted once in that category (e.g., musculoskeletal pain for patients reporting arthralgia and low back pain). Hierarchical clustering was used to group the different categories, according to the frequency of their co-occurrence in the individual patient data. Hierarchical clustering has already been described as useful in pain studies to identify profiles, multiple associations of descriptors, and combine symptoms into subgroups [7]. The final analysis was expressed by a network plot using Ward coupling, i.e., showing the clusters with the smallest variance (the most similar) and a dendrogram (Appendix A).

## 3. Results

### 3.1. Patient Characteristics and Quantitative Analyses

Patients had a mean age of 55.7 years (SD 14) ranging from 28 to 85. All were women except for 1 man from the ketorolac group. All patients received the assigned treatment, and all underwent surgery after randomisation (Table 1, Figure 1) (adapted from [3]).

No patient was taking strong opioids, and only one patient was taking weak opioids preoperatively and stopped after one year. After one year, one patient took weak and one took strong opioids. After two years, three patients took weak opioids.

### 3.2. Pain Characteristics

As previously indicated, there was no difference, between the ketorolac and the placebo group, in pain on D1 after surgery, neither at rest (*p* = 0.620) nor in movement (*p* = 0.254) (Figure 2), even if a difference of 3 mg of morphine equivalent appeared (ketorolac group: 5 mg (interquartile range 25–75: 0–10); placebo group: 8 mg (4–14); *p* = 0.001) (Table 2). No differences appeared at other time points neither for acute pain, nor for chronic pain (Table 2). Changes in pain scores were compared between the groups, not revealing any difference.

In terms of persistent pain, most patients described pain after one and two years. Most of them (67%) reported permanent pain at one year, but not at 2 years (45%) (Table 2, Figure 3 and Figure 4). The largest category among patients describing pain was musculoskeletal pain, and the smallest was pain associated with polyneuropathy (at 1 and 2 years, 23 and 20, with, respectively, 17 and 15 having received a chemotherapy, *p* = 0.004 and 0.01, when compared with no chemotherapy). Permanent pain (pain at all times) was mainly described in patients with musculoskeletal pain at one and two years, although pain associated with polyneuropathy was mainly permanent during the first year (Table 3, Figure 3).

In terms of pain permanence and location, approximately half of the patients moved from one category to another between the first and the second year. For example, 39% of patients (37/94) with permanent pain also described it as permanent in the second year, while 18% (11/60) of the pain-free patients of the first year reported permanent pain during the second year. Only 44% of patients (89/203) remained in the same category for the permanence of pain. A similar pattern was observed for localisation. In other words, 30% of the patients (11/37) reporting pain in the surgical area after one year described it as such after two years. In total, only 48% (98/203) remained in the same category (Figure 4 and Figure 5).

### 3.3. Similarities and Differences in Patients with Pain after One and Two Years

No significant difference between patients with and without pain after 1 and after 2 years was detected with regards to age, BMI, type of surgery, description of preoperative, and acute postoperative pain (Table 4). Any differences observed after 1 year in terms of age and BMI (both higher) statistically disappeared after Bonferroni’s correction for the significance of alpha (0.0025).

Pain trajectories were explored but did not differ between patients with or without persistent pain (respectively, median pain score on a simple verbal scale (0 to 4) at movement, 3 days after surgery: 1 (0–1) vs. 0 (0–1); *p* = 0.61 after 1 year and 0 (0–1) vs. 1 (0–1); *p* = 0.47 after two years).

## 4. Discussion

### 4.1. Summary of the Results

This study shows that a single dose of ketorolac induced no detectable change in terms of pain scores (even if associated with a modest decrease in opioids at D1 in the ketorolac group: 5 mg (interquartile range 25–75: 0–10); compared to the placebo group: 8 mg (4–14); *p* = 0.001). More interestingly, the pain trajectories were explored here, revealing no difference between the groups, but clearly showing that most patients described a postoperative appearance of pain in areas other than the surgical site, with musculoskeletal pain being the most important group. The description of pain changed in most patients during the second postoperative year, i.e., passed from one category to another (no pain, permanent, or non-permanent pain, but also, the localisation). This phenomenon includes patients without pain at one year. Persistent pain was therefore more frequently musculoskeletal, rather than associated with polyneuropathy, although the latter treatment was clearly a risk factor for permanent pain (*p* = 0.004 and 0.01 at 1 and 2 years, compared to the absence of chemotherapy). The type of surgery and other types of adjuvant treatment(s) were not associated with pain outcomes here.

Were these results surprising? For its anticancer effect, a single dose of ketorolac was not sufficient to induce a difference in terms of oncological results. One suspected reason is that the use (of a single dose) may have been too short [3]. This may also be true for the analgesic effect, as the use of non-steroidal anti-inflammatory drugs (NSAIDs) is generally recommended for the duration of postoperative pain [8]. This lack of effect from a single dose of ketorolac, except a small opioid-sparing effect, was thus not very surprising. However, the pain patterns described were not expected. Gartner et al. have developed one of the most used questionnaires for assessing pain after surgery for breast cancer and report a 40% incidence of pain in areas other than the surgical site [9], which is consistent with our results. New research in the field has focused on an intervention to reduce the incidence of breast (or surgical) pain after breast cancer surgery. In these studies, pain in other parts of the body has generally not been reported and it remains unclear if questions about pain beyond the surgical site were used [10,11,12,13]. However, musculoskeletal pain, such as shoulder pain, has been recognised as a significant burden after breast surgery [2].

Taken together, our results and previous ones suggest that the type of surgery might not be so important in breast cancer surgery with respect to the risk of persistent pain [9,13]. However, one would have expected to see adjuvant treatments associated with certain types of pain, such as aromatase inhibitors, with arthralgia, which was not the case here [14]. This study, however, was not designed for and would clearly lack power and underfitting for the construction of a multivariable model. These results must therefore be considered as generating hypotheses. Even if only partially and in a specific dataset, our results challenge the existing theoretical framework proposing that the persistent pain after breast cancer surgery is mainly located in the operating site and with constant characteristics over time. In other words, permanent pain after breast surgery may not be (or not primarily) directly caused by the surgical insult and even related to the surgery. A late onset of permanent pain has been observed here and it seems that how patients experience permanent pain and how their experience changes (or is influenced by interfering factors) during the postoperative years is mostly unknown. Apart from the type of surgery discussed above and adjuvant treatments, this may be related to the inherent heterogeneity of the patient profile. This opens up a new area of research, trying to implement better patient-centred approaches, including in research projects. The patients’ perspective on their own goals and what is acceptable and important, in terms of pain and functional goals, is essential to understanding important areas of research. This is also true for the treatments and strategies which should be developed, in a more holistic way, because pain clearly cannot be considered as an isolated phenomenon, in time or in the body. This should make us consider that the importance of neurophysiological mechanisms in persistent pain is real but not necessarily the most important, at least in the long term. Thus, the remaining issues need to be considered from a holistic perspective when focusing on persistent pain. This is also true of the intriguing late onset of pain in the surgical area, which is possible but difficult to explain. Is this because women who did not have pain may return earlier to activities that may cause pain later, e.g., starting heavy lifting earlier in their recovery, which had implications for longer term pain and recovery? This could explain the lack of relationship observed between the trajectories of acute pain and the occurrence of persistent pain. Or is it the opposite? One could also argue the need to explore the impact of visibility/size of the incision and bruising on pain perception in the short and longer term. Late onset pain merits further study, given the possibilities of preventive interventions, including the potential influence of a multidisciplinary approach.

### 4.2. Limitations

These observations should not be overinterpreted. These are limited to one specific dataset, with its inherent limits in terms of precision and possibly questionable generalisability, given the high oncological risk profile of the included patients. The risk of type I error linked to the multiplicity of analyses may have been controlled too conservatively with Bonferroni’s correction, making it impossible to conclude on the effect of age and BMI on pain a year after surgery. Since a young age is mainly associated with a higher risk of persistent pain, it would have been unexpected that the patient without pain, one year after surgery, was significantly younger than the patients describing pain [1]. Finally, we were unable to confirm the role of endocrine therapy, radiotherapy, and chemotherapy here, but the study was not designed for this and was underpowered, even if the effect of these treatments can without any doubt be complicated by the development of pain [14]. In future studies, it is necessary to integrate other factors that may influence pain during long-term follow-up such as hormonal therapies, other hormonal symptoms (hot flashes and night sweats, mood swings), lymphedema and extension of lymph node dissection, concomitant medical conditions (like arthritis and diabetes), psychological and psychiatric factors, other illnesses, and social factors.

## 5. Conclusions

Pain is a complex phenomenon, but also a fragile (from a methodological point of view) and unstable (from a clinical perspective) endpoint. Pain after breast cancer surgery does not necessarily mean breast pain, and musculoskeletal pain and other pains may represent a heavier burden. The permanence of pain and the pain phenotype can change dramatically over time.

## Figures and Tables

**Figure 1 jcm-09-03831-f001:**
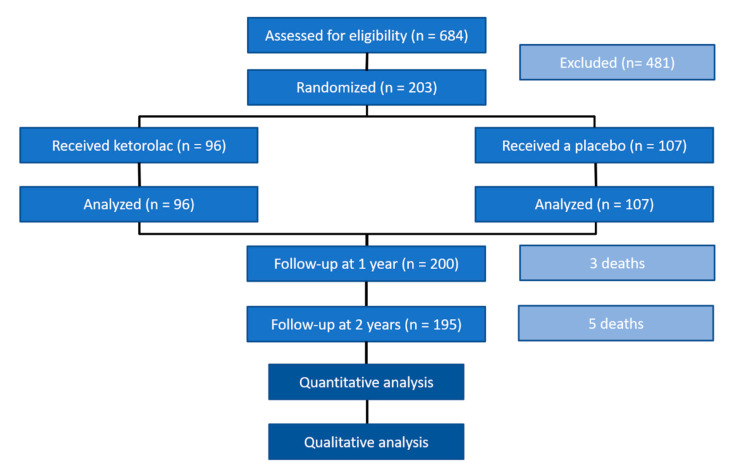
Study flow chart.

**Figure 2 jcm-09-03831-f002:**
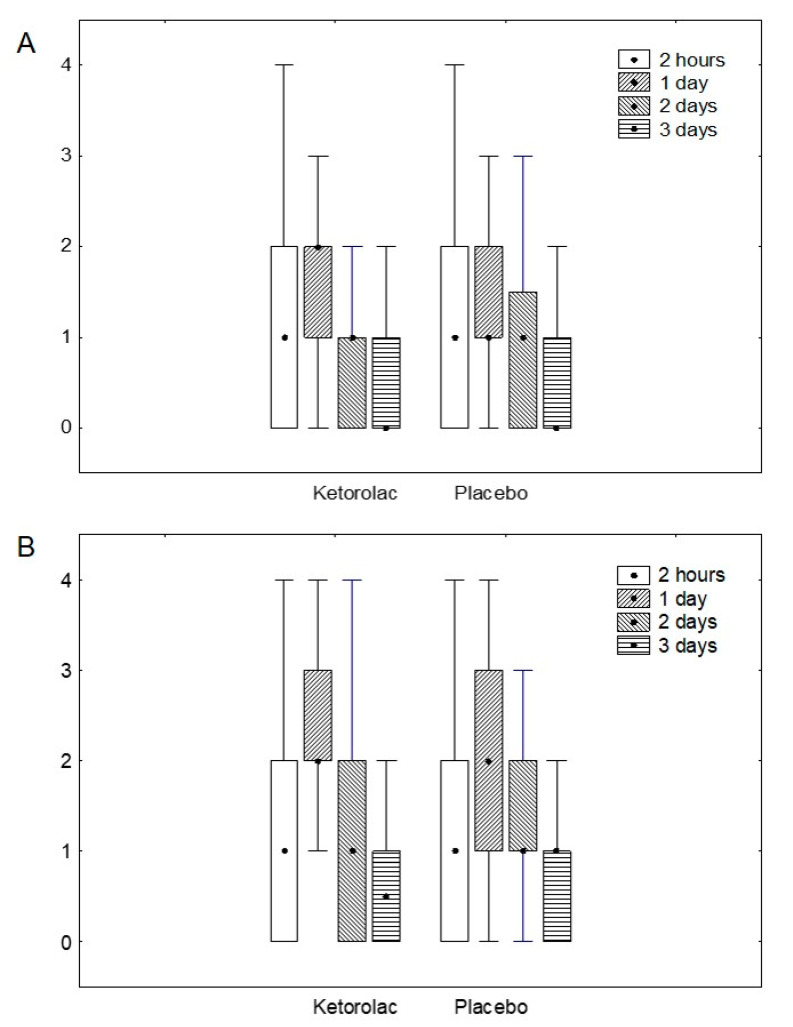
Acute pain score on a simple verbal scale (0 to 4) at rest (**A**) and movement (**B**) after surgery. *p* > 0.05 for all the comparisons. Pain scores are expressed as median, interquartile range (25–75), and range.

**Figure 3 jcm-09-03831-f003:**
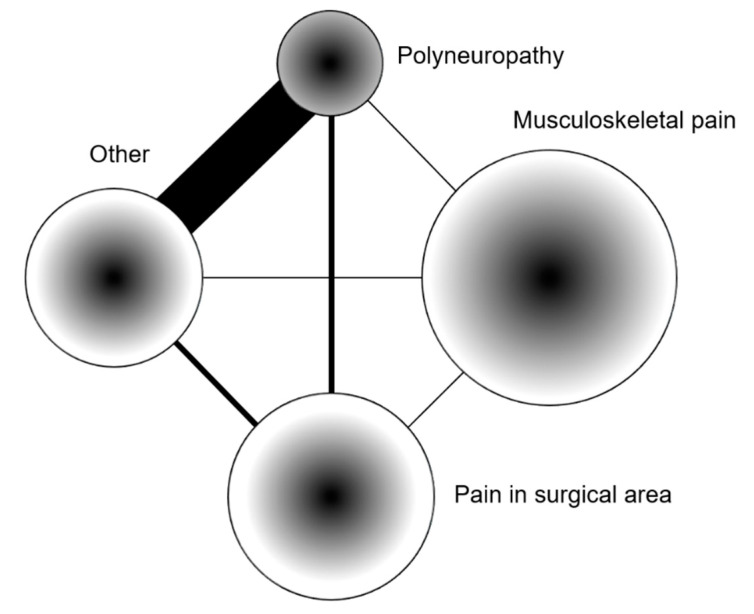
Network plot reporting the numbers of patients reporting pain in different categories (size), the proportion of reported pain permanence (gradient: black—permanent; white—non-permanent), and the frequency of their co-occurrence (lines thickness indicates the closeness of the relationship, based on the inverse Ward distance).

**Figure 4 jcm-09-03831-f004:**
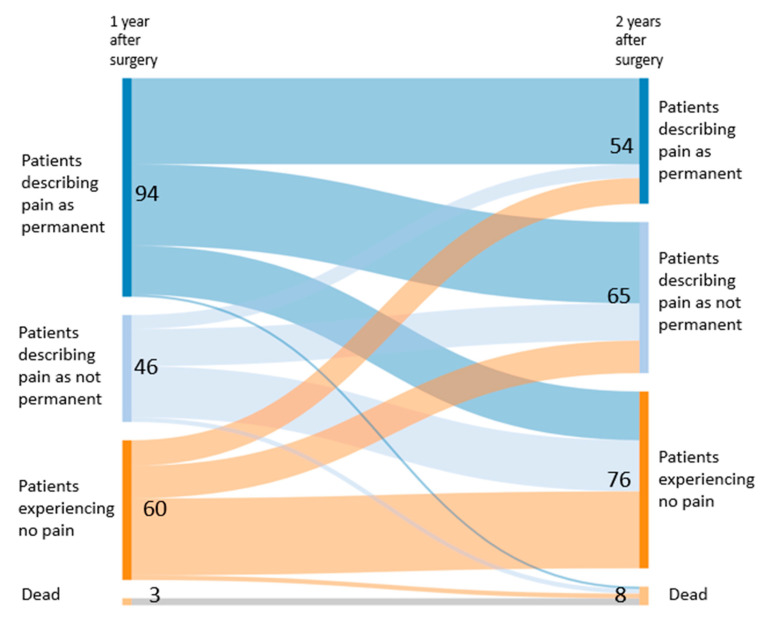
Pain permanence trajectories between the first and the second postoperative year.

**Figure 5 jcm-09-03831-f005:**
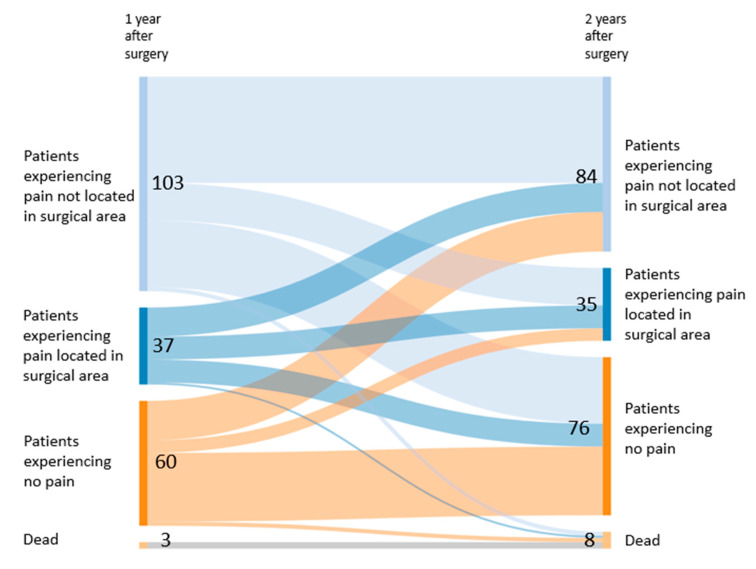
Pain localisation trajectories between the first and the second postoperative year.

**Table 1 jcm-09-03831-t001:** Baseline characteristics of the patients, tumours, and treatments (adapted from Forget 2019).

Characteristics	Entire Series *n* = 203	Ketorolac *n* = 96	Placebo *n* = 107
Age, years			
Mean (SD)	55.7 (14.0)	56.1 (14.0)	55.4 (13.9)
Range	28–85	30–85	28–85
Gender, *n* (%)			
Female	202 (99%)	95 (99%)	107 (100%)
Male	1 (1%)	1 (1%)	0
Chemotherapy, *n* (%)			
Yes	161 (79%)	80 (83%)	81 (76%)
No	42 (21%)	16 (17%)	26 (24%)
If chemotherapy (anthracyclines with/without taxanes), type (%)			
Adjuvant	54 (27%)	27 (28%)	27 (25%)
Neoadjuvant	107 (53%)	53 (55%)	54 (51%)
Type of surgery, *n* (%)			
Mastectomy	119 (59%)	60 (63%)	59 (55%)
Breast-conserving surgery	82 (40%)	34 (35%)	48 (45%)
Missing	2 (1%)	2 (2%)	0
Type of lymphadenectomy, *n* (%)			
None	14 (7%)	6 (6%)	8 (8%)
Sentinel	12 (6%)	5 (5%)	7 (6%)
Complete axillary	176 (87%)	85 (89%)	91 (85%)
Missing	1 (1%)	0	1 (1%)
Post-operative radiotherapy, *n* (%)			
Yes	165 (81%)	77 (80%)	88 (82%)
No	38 (19%)	19 (20%)	19 (18%)
Endocrine therapy, *n* (%)			
Yes	138 (68%)	69 (72%)	69 (64%)
No	64 (31%)	27 (28%)	37 (35%)
Missing	1 (1%)	0	1 (1%)

**Table 2 jcm-09-03831-t002:** Preoperative, acute, and persistence postoperative pain.

Characteristics	Ketorolac (*n* = 96)		Placebo (*n* = 107)		*p*-Value
Preoperative chronic pain					
	13	(14%)	24	(23%)	0.356
Preoperative pain (any type, including post-biopsy)					
Every day	59	(62%)	62	(58%)	0.583
One to 3 days a week	0	(0%)	1	(1%)	
Less than once a week	27	(28%)	36	(34%)	
Missing	10	(10%)	8	(8%)	
Acute pain					
Pain at rest at D1 after surgery Verbal simple scale (0–4), *N* (%)					0.620
0	18	(19%)	17	(16%)	
1	35	(37%)	31	(29%)	
2	37	(38%)	47	(44%)	
3	5	(5%)	10	(9%)	
4	1	(1%)	1	(1%)	
Missing	0	(0%)	1	(1%)	
Morphine equivalent at D1 (in mg) Median (interquartile range 25–75)	5	(0–10)	8	(4–14)	0.001
Mean change in pain score to Day 1 (SD)	−0.6	(1.16)	−0.7	(1.24)	0.357
Mean change in pain score to Day 2 (SD)	−0.1	(1.42)	0.1	(1.42)	0.631
Mean change in pain score to Day 3 (SD)	0.7	(1.24)	0.8	(1.27)	0.869
Persistent pain 1 year after surgery					
Presence of any type of pain	72	(75%)	70	(65%)	0.137
Average pain score Mean Visual Analogue Scale (0–10) (SD)	2.6	(1.4)	2.9	(1.4)	0.166
Persistent pain 2 years after surgery					
Presence of any type of pain	57	(63%)	62	(59%)	0.541
Average pain score Mean Visual Analogue Scale (0–10) (SD)	2.4	(1.5)	2.3	(1.7)	0.805

**Table 3 jcm-09-03831-t003:** Localisation and permanence of pain 1 and 2 years after surgery.

**Localisation of Pain after 1 Year**	***n***	**Experiencing Permanent Pain (%)**
Pain in surgical area	37	21 (58)
Polyneuropathy	23	22 (96)
Musculoskeletal	56	37 (66)
Other	28	17 (61)
None	61	0
Total: 200 patients (with and without pain), 205 reported pain types		
**Localisation of Pain after 2 Year**	***n***	**Experiencing Permanent Pain (%)**
Pain in surgical area	35	18 (51)
Polyneuropathy	20	10 (50)
Musculoskeletal	66	31 (47)
Other	60	27 (45)
None	77	0
Total: 195 patients (with and without pain), 258 reported pain types		

**Table 4 jcm-09-03831-t004:** Patients’ characteristics, with and without pain, 1 and 2 years after surgery. No statistically significant difference after Bonferroni’s correction for multiple comparisons. *p*-values according to Chi square tests and Fisher exact tests as appropriate.

Patients’ Characteristics	1 Year after Surgery			2 Years after Surgery		
	Pain *n* = 140	Pain Free *n* = 60	*p*	Pain *n* = 119	Pain Free *n* = 76	*p*
Gender n (%)						
Female	139 (99)	60 (100)	-	118 (99)	76 (100)	-
Male	1 (1)	0	-	1 (1)	0	-
Age (Mean; SD; CI)	59.7 (26.5; 55.2–65.1)	52.8 (13.3; 49.3–56.2)	0.02	56.0 (13.8; 53.3–63.6)	58.5 (13.8; 52.8–59.1)	0.87
BMI (Mean; SD; 95%CI)	26.0 (5.0; 25.1–26.8)	24.4 (4.4; 23.2–25.5)	0.03	25.8 (5.3; 24.8–26.8)	24.8 (4.1; 23.8–25.7)	0.14
Preoperative Pain (Yes/No)	28/108	6/52	0.13	23/91	10/65	0.31
Surgery *n* (%)			0.53			0.50
Mastectomy	84 (60)	32 (53)		67 (56)	46 (61)	
Quadrantectomy/Lumpectomy	55 (39)	27 (45)		52 (44)	28 (37)	
Bilateral	13 (9)	8 (13)		11 (9)	10 (13)	
Axillary lymph node resection *n* (%)			0.42			0.69
Complete axillary lymph nodes resection	119(85)	54 (90)		104(87)	64 (84)	
Sentinel lymph node	8 (6)	4 (7)		6 (5)	6 (8)	
None	12 (9)	2 (3)		8 (7)	6 (8)	
Hormonotherapy	96 (69)	43 (72)	0.66	47 (41)	37 (49)	0.30
Sensation of phantom breast (%)	1 (1)	0	-	0	0	-
Association with pain (Yes/No)	0/1	0/0	-	0/0	0/0	-

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
