# Peer review of "Characterization of Preoperative, Postsurgical, Acute and Chronic Pain in High Risk Breast Cancer Patients"

_jcm, 2020, doi:10.3390/jcm9123831_

Round 1
Reviewer 1 Report
It is a very interesting study but it has major flaws to the presentation of the results. Specifically, the authors present a two- arm study, but after the 1st Table, they present the results at the total of the patients. They should present the results between the two groups from the beginning to the end of the follow-up. The study is comparative. Also, at Table 1. they have not added the p-value.
Also, they should add a table with the analgesics that patients have recieved.
Furthermore, the authors should discuss their results more.
Reviewer 2 Report
good work.
suggestions for its improvements:
- make it clearer in the abstract that this is a KBC sub-study: lines 36-37 refere to KBC. so wtitten, there are possible misunderstandings about this substudy. i.e.: line 36 avoid "which" and use ". KBC was..."
- inclusions criteria line 87-88: if the patients are as indicated, they are all at high risk of recurrence. in this case you must indicate it in the title "... in high risk breast cancer patients"..
- line 184 "is"?
- NSAIDs (better indicate the meaning even well-known)
- moreover, the weak point in this paper is the discussion. go deeper in your analys, take into account as possible limitations the eterogeneity of the patients in terms of differents surgery, different adiuvant treatment. please cite other possible approaches in the post-surgical control such as homeopathy and acupuncture. i.e. in Bosco F, Cidin S, Maceri F, Ghilli M, Roncella M, De Simone L. An integrated approach with homeopathic medicine and electro-acupuncture in anaesthesiology during breast cancer surgery: Case reports. J Pharmacopuncture. 2018 Jun;21(2):126-131. doi: 10.3831/KPI.2018.21.016. Epub 2018 Jun 30. PMID: 30151314; PMCID: PMC6054085.
- it would have been better to limit the sample to a uniform group of surgeries: mastectomy or BCS, and to a uniform group of post-operative therapies because they are closely related to post-operative pain, particularly after 1 or 2 years.
Round 2
Reviewer 1 Report
Major improvent of the manuscript. But the authors should make a minor change in Table 3 and 4 , and Figure 4 and 5, so as to show the comparison between the two groups, not the results in the total of patients.
Author Response
We thank the reviewer for the suggestion. However, according to the ICH-e9 guidelines for the communication of clinical trial results (https://www.ema.europa.eu/en/ich-e9-statistical-principles-clinical-trials), we are not allowed to make inferential analyzes based on explanatory analyzes. In other words, the analyzes presented in the tables mentioned are justified to explain the results observed in general, but we cannot go further, not only because the proposed modifications were not pre-planned and the power of the study did not take them into account. , but also because they would violate the assumptions of the aforementioned statistical principles and recommendations. We hope the reviewer understands these constraints.